# Towards Development of a Non-Toxigenic *Clostridioides difficile* Oral Spore Vaccine against Toxigenic *C. difficile*

**DOI:** 10.3390/pharmaceutics14051086

**Published:** 2022-05-19

**Authors:** Jaime Hughes, Carl Aston, Michelle L. Kelly, Ruth Griffin

**Affiliations:** 1Synthetic Biology Research Centre, School of Life Sciences, The University of Nottingham Biodiscovery Institute, Nottingham NG7 2RD, UK; jaime.hughes@nottingham.ac.uk (J.H.); carl.aston@nottingham.ac.uk (C.A.); michelle.kelly@nottingham.ac.uk (M.L.K.); 2NIHR Nottingham Biomedical Research Centre, Nottingham University Hospitals NHS Trust, The University of Nottingham, Nottingham NG7 2UH, UK

**Keywords:** *Clostridioides difficile*, oral vaccines, mucosal, sIgA

## Abstract

*Clostridioides difficile* is an opportunistic gut pathogen which causes severe colitis, leading to significant morbidity and mortality due to its toxins, TcdA and TcdB. Two intra-muscular toxoid vaccines entered Phase III trials and strongly induced toxin-neutralising antibodies systemically but failed to provide local protection in the colon from primary *C. difficile* infection (CDI). Alternatively, by immunising orally, the ileum (main immune inductive site) can be directly targeted to confer protection in the large intestine. The gut commensal, non-toxigenic *C. difficile* (NTCD) was previously tested in animal models as an oral vaccine for natural delivery of an engineered toxin chimera to the small intestine and successfully induced toxin-neutralising antibodies. We investigated whether NTCD could be further exploited to induce antibodies that block the adherence of *C. difficile* to epithelial cells to target the first stage of pathogenesis. In NTCD strain T7, the colonisation factor, CD0873, and a domain of TcdB were overexpressed. Following oral immunisation of hamsters with spores of recombinant strain, T7-0873 or T7-TcdB, intestinal and systemic responses were investigated. Vaccination with T7-0873 successfully induced intestinal antibodies that significantly reduced adhesion of toxigenic *C. difficile* to Caco-2 cells, and these responses were mirrored in sera. Additional engineering of NTCD is now warranted to further develop this vaccine.

## 1. Introduction

*Clostridioides difficile* is a Gram-positive, spore-forming anaerobe found in the gastrointestinal (GI) tract of humans and animals and is the leading cause of nosocomial diarrhoea worldwide. Up to 17.5% of healthy adults globally are colonised with this organism and display no disease symptoms [1,2,3], as the gut microbiome provides an effective colonisation resistance barrier against *C. difficile* [4,5]. When the status of the microbiome is compromised for example due to antibiotic exposure, susceptibility to *C. difficile* infection (CDI) is induced [6,7]. Clinical manifestations range from mild diarrhoea to colitis, toxic megacolon and potentially death [8]. There are few antimicrobials available to treat CDI (Vancomycin, Fidaxomicin and Metronidazole) [9] which have limited efficacy and as such, *C. difficile* poses a serious health threat classified as “urgent” by the Centers for Disease Control and Prevention (CDC) and “critical” by the World Health Organisation. The global CDI incidence rate ranges from 1.1 to 631.8 per 100,000 population per year [10], and the mortality rate directly attributed to CDI is estimated at 5% [11,12].

CDI is transmitted as a highly contagious spore by the faecal–oral route [13,14]. Spores are resistant to the harsh conditions of the stomach [15] but readily germinate in the ileum in the presence of cholate-derived germinants [16,17,18]. The primary bile acid, cholic acid, is secreted into the duodenum and is conjugated with taurine or glycine to produce taurocholate or glycocholate. Enabled by certain species of the intestinal microbiota, bile salts are deconjugated in the ileum to cholate, and in the large intestine, converted to secondary bile acids. In the absence of these species following their elimination by broad spectrum antibiotics, conversion to secondary bile acids is reduced, resulting in increased concentration of germinants and leading to increased susceptibility to CDI [19,20,21]. Vegetative cells from the outgrowth of germinated spores colonise the colon via adherence and motility factors such as Cell Wall Protein Cwp66, Surface Layer Proteins (Slps) and the protease Cwp84 (which cleaves the Slp precursor SLpA) [22,23], colonisation factor CD0873 [24,25], heat shock protein GroEL and flagella [26,27,28]. Two large toxins encoded in the pathogenicity locus (paLoc), TcdA (308 kDa) and TcdB (270 kDa) are responsible for the pathology characteristic of CDI [29]. These exotoxins enter epithelial cells and glycosylate and inactivate Rho GTPases affecting the cytoskeleton, which results in a breakdown of gut barrier integrity and loss of functionality [8]. The life cycle of *C. difficile* completes when vegetative cells sporulate. Spores exit the body via diarrheal shedding and transmit to other hosts.

CDI is difficult to treat because the antibiotics used to treat infection exacerbate dysbiosis. Since antibiotics are only active against vegetative cells, once the course is completed, residual spores germinate in the conducive environment of the disrupted microbiota, leading to recurrent CDI. After effective treatment of the first episode, at least one recurrent episode occurs in 15–35% of patients, and a further episode occurs in 33–65% of patients who have had two or more infections [30,31,32].

With the limitation of available treatment options, attention has turned to developing vaccines as a logical, cost-effective approach to prevent CDI. Several studies demonstrated that anti-toxin serum antibodies correlate with natural protection [33,34]. Inactivated toxins or components of toxins became choice vaccine candidates to test in clinically relevant models, hamsters and mice, with administration predominantly parenteral. Two intramuscular toxoid vaccines progressed to human trials: formalin-inactivated toxins (Sanofi Pasteur) and genetically and chemically inactivated toxins (Pfizer). Phase I and II trials demonstrated that the vaccines are safe and strongly induce serum antibodies that neutralise the toxins [35,36]. Phase III tested the efficacy of these vaccines in individuals 50 years or older who were at risk of CDI; however, both vaccines failed to prevent primary CDI [37,38].

For gut pathogens, in particular those that are non-invasive such as *C. difficile*, immunising orally to target the ileum can drive the local protection needed and thus potentially prevent disease. At mucosal surfaces, secretory IgA (sIgA) antibodies serve as the first line of defence by agglutinating pathogens and antigens and by preventing their interaction with the epithelium. As well as playing this vital innate role known as immune exclusion, sIgA is an immunopotentiator, i.e., promotes an adaptive response. sIgA binds to the antigen and facilitates its transcytosis to underlying dendritic cells (DC) in the gut-associated lymphoid tissue (GALT) [39]. Ultimately, B cells are activated and mature to become IgA+ plasma cells, releasing sIgA into the gut lumen [40]. The clear relationship between sIgA and CDI was highlighted in two studies, which showed that low total faecal IgA levels and low total colonic IgA-producing cells correlated with prolonged CDI symptoms and higher rates of recurrence [41,42].

A major bottleneck, however, in developing oral vaccines is antigen degradation in the stomach. A potential solution for the delivery of intact antigens to the ileum is the deployment of a gut commensal spore-forming organism such as non-toxigenic *C. difficile* (NTCD). Spores naturally resist the harsh conditions of the stomach and germinate in the small intestine into metabolically active vegetative cells where they can release engineered antigens.

Wild-type NTCD strain M3 has already been tested in humans as a biotherapeutic against toxigenic *C. difficile* (TCD) [31]. In a Phase II trial, M3 was investigated for the prevention of recurrent CDI in patients experiencing their first episode or first recurrence, who had received antibiotic treatment for CDI. A significant decrease in recurrence rate was shown, which was associated with colonisation by M3 [43] that was potentially enabled by antibiotic-induced gut dysbiosis. The purpose of deploying NTCD in this study was not to promote colonisation resistance by competition but was solely to deliver engineered antigens to the small intestine to induce an adaptive humoral response in non-antibiotic treated animals. Our ultimate goal will be to vaccinate individuals approaching “at risk” of CDI who are not taking antibiotics, such as those over 65 years and entering long-term stay at hospitals or nursing homes or patients awaiting elective surgery who will require antibiotics. A previous study showed that NTCD can be engineered to induce antibodies directed against TCD. The NTCD strain chosen expressed a plasmid encoding a chimeric protein comprising the glucosyltransferase and cysteine proteinase domains of TcdB fused with the receptor binding domain (RBD) of TcdA [44]. Mice and hamsters were immunised orally with spores of the recombinant strain, NTCD_mTcd138, and faecal and systemic IgA and IgG responses were investigated. Mice generated toxin-neutralising antibodies and were fully protected from CDI, while hamsters induced lower titres of toxin-neutralising serum antibodies and were significantly but not fully protected [44].

Building on this study, we set out to investigate whether recombinant NTCD could be further exploited to induce antibodies with colonisation-blocking properties to potentially prevent the first stage of pathogenesis. NTCD T7 was chosen, as this strain was previously characterised for its biotherapeutic properties against CDI in the hamster model [45,46]. The colonisation factor, CD0873, was chosen as we showed that it induces colonisation blocking antibodies in hamsters when given as recombinant protein orally [25,47]. The receptor binding domain (RBD) of TcdB was chosen, as it is known to be immunogenic and to confer full protection in mice [48]. The antigens were cloned from TCD strain 630 and expressed under the strong constitutive *fdx* promoter of *Clostridium sporogenes* episomally to create two recombinant strains, T7-0873 and T7-TcdB.

Following oral immunisations with 10^6^ spores of either recombinant strain, intestinal fluid and sera were investigated for functional antibody responses. T7-0873 elicited significant titres of intestinal antibodies with significant adherence blocking activity, and these responses were mirrored by serum IgG. T7-TcdB also induced TcdB-specific antibodies with adherence-blocking activity but in serum only and with no toxin-neutralising activity. To conclude, we show that NTCD engineered to express CD0873 is highly effective at inducing adherence-blocking antibodies both locally and systemically. Developing the NTCD vaccine by modifications of its chromosome to reduce its persistence in the host and to express elevated levels of antigens is required next for the generation of a novel prophylactic oral spore vaccine against *C. difficile*.

## 2. Materials and Methods

### 2.1. Bacterial Strains

*Escherichia coli* TOP10 (Invitrogen, Paisley, UK) was used for the purpose of cloning. The *E. coli* conjugal donor strain, sExpress, was used for plasmid transfer into non-toxigenic *Clostridioides difficile* (NTCD) [49]. *E. coli* strains were cultured aerobically in Luria Bertani (LB) broth with shaking or on LB agar (Fisher Bioreagents, Loughborough, UK) at 37 °C. Where appropriate, Chloramphenicol (Sigma, Merck Group, Feltam, UK) was added to a final concentration of 25 µg/mL.

Toxigenic *C. difficile* (TCD) strain 630 was obtained from ATCC (Manassas, VA, USA). Non-toxigenic *C. difficile* (NTCD) strain T7 is part of the SBRC culture collection and was provided by the Edward Hines Jr Veterans Affairs Hospital (5000 5th Avenue, Hines, IL 60141, USA: Investigator Dale Gerding). T7 was originally isolated from asymptomatic hospitalised patients and found to occur at high frequency [46]. T7 and derived strains were cultured in Brain Heart Infusion (BHI) broth or on BHI agar (Oxoid, Basingstoke, UK) supplemented with 0.5% yeast extract (Oxoid, Basingstoke, UK) and 0.1% cysteine (ThermoFisher Scientific, Loughborough, UK) (BHIS) containing 250 µg/mL D-cycloserine and 8 µg/mL cefoxitin (Oxoid, Basingstoke, UK) (BHIS CC). The strains were incubated overnight at 37 °C in an anaerobic workstation (Don Whitley Scientific, Bingley, UK) with an atmosphere of CO_2_ (10%), H_2_ (10%) and N_2_ (80%). Media were pre-reduced before use. Where appropriate, Thiamphenicol (Acros Organics, Geel, Belgium) was added to a final concentration of 15 μg/mL.

### 2.2. Preparation of Genomic DNA

Genomic DNA was isolated from NTCD T7 for Illumina sequencing and from TCD 630 for PCR amplification of antigens by phenol–chloroform extraction. Then, 10 mL of overnight BHIS CC culture were centrifuged and the pellet resuspended in 180 μL of lysis buffer (PBS containing 45 mg/mL lysozyme), and then incubated at 37 °C for 1 h with gentle agitation. Next, 20 μL RNase (Sigma, Merck Group, Feltham, UK) were added and incubated at room temperature for 10 min. Then, 25 μL proteinase K (Qiagen, Venlo, Netherlands), 85 μL ddH_2_O, and 110 μL 10% SDS solution were added, mixed by inversion and incubated at 65 °C for 30 min with gentle agitation. An equal volume of Phenol/Chloroform/Isoamyl alcohol (PCI) (25:24:1) (Sigma, Merck Group, Feltham, UK) (pH8) was added and mixed thoroughly by inversion and then transferred to a 5PRIME Phase Lock Gel (PLG) tube (Quantabio, Beverly, MA, USA) and centrifuged at 16,000× *g* for 5 min. The upper fraction was removed and added to a fresh Phase Lock tube, and the phenol–chloroform extraction was repeated 2 more times. The upper phase was transferred to an Eppendorf containing 0.1 volume of 3 M NaAc and 2.0 volume of ice-cold 100% EtOH and mixed by inversion and then left at −80 °C for 1 h. The sample was centrifuged at maximum speed for 15 min at 4 °C. The supernatant was removed and the pellet centrifuged for a further 3 min, residual EtOH removed and the DNA pellet air-dried for 45 min at room temperature before resuspension in 50 μL ddH_2_O.

### 2.3. Illumina Sequencing

DNA concentrations were measured using the Qubit Fluorometer and the Qubit dsDNA BR Assay Kit (ThermoFisher Scientific, Loughborough, UK) and 250 ng were used for sequencing library preparation. Indexed sequencing libraries were prepared using the Nextera DNA Flex Library Prep Kit (Illumina, San Diego, CA, USA) and Nextera DNA CD Indexes (Illumina, San Diego, CA, USA). Libraries were quantified using the Qubit Fluorometer and the Qubit dsDNA HS Kit (ThermoFisher Scientific, Loughborough, UK). Library fragment-length distributions were assessed using the Agilent TapeStation 4200 and the Agilent High Sensitivity D1000 ScreenTape Assay (Agilent, Santa Clara, CA, USA). Libraries were pooled in equimolar amounts and final library quantification performed using the KAPA Library Quantification Kit for Illumina (Roche, Basel, Switzerland). The library pool was sequenced on an Illumina MiSeq using the MiSeq Reagent Kit v2 (500 cycle) (Illumina, San Diego, CA, USA) to generate 250 bp paired-end reads. The whole genome shotgun was deposited in GenBank as a BioProject under Accession PRJNA826427.

### 2.4. Bioinformatics Analysis

*C. difficile* vaccine candidates were first identified from the literature and the relevant genes of TCD strain 630 retrieved from NCBI (https://www.ncbi.nlm.nih.gov (accessed on 15 April 2022)). Using these genes as query sequences, nucleotide BLAST (Basic Local Alignment Search Tool) (BLASTn) analyses were conducted against the whole-genome shotgun of NTCD T7. Translated sequences of the genes identified in strain 630 were used as query sequences in protein BLAST (BLASTp) analyses against the translated shotgun of T7. BLASTn analyses were conducted using query sequences of strain 630 to locate the integration site in strain T7.

### 2.5. NTCD Spore Preparation

After incubation of *C. difficile* strains on BHIS CC plates for 5 days, the culture was scraped and resuspended in 1 mL PBS. The suspension was heated at 60 °C for 30 min to kill vegetative cells and then centrifuged at 12,000× *g* for 1 min. Pellets were washed 3 times in dH_2_O and then resuspended in 1 mL sterile dH_2_O and stored at −80 °C. Spores were enumerated by plating 1:10 serial dilutions onto BHIS CC containing 0.1% taurocholate (ThermoFisher Scientific, Loughborough, UK) and counting colony forming units (CFU) from germinated vegetative cells after overnight incubation.

### 2.6. Ex Vivo Spore Germination Assay

The small intestine of 6 hamsters was collected and cut into 3 equal length portions. Proximal, mid and distal regions of each hamster were separately combined. After cutting each portion longitudinally and vortexing, the tissues were centrifuged at 9000× *g* for 10 min. The supernatant was filter sterilised through a 0.22 µm filter to remove any commensal bacteria present. Then, 45 µL of the fluid were incubated with 5 µL of T7 spore suspension (4 × 10^5^ spores per mL) for 1 h under anaerobic conditions. As a positive control, 5 µL of the spore suspension were incubated in BHIS containing 0.1% taurocholate. Next, 4 × 5 µL of 1:10 serial dilutions were plated onto BHIS CC agar containing Kanamycin (50 µg/mL). All plates were incubated at 37 °C overnight anaerobically. CFU were enumerated and compared with that of the positive control (defined as 100% germination).

### 2.7. Cloning of Antigens in pMTL84123 for Expression in T7

Given our previous finding that the expression of heterologous genes can be aided by the inclusion of the native promoter, in addition to the plasmid promoter [50], primers were designed to amplify sequence from genomic DNA of TCD 630 from −83 nucleotides upstream of the start codon to the end of the stop codon of CD0873. The putative promoter region was identified from −67 to −22 nucleotides upstream of the start codon using Promoter Prediction by Neural Network programme, applicable for prokaryotes. The primers used were “CD0873-SacI-for”, 5′-GCGGCGGAGCTCTTACAATAATTATGGTTA-3′ and “CD0873-XbaI-rev”, 5′-GCCGCCTCTAGACTATTCTTGTTTAGTCTTTA-3′. The region encoding TcdB-RBD was amplified using forward primer, “TcdB-XbaI-for”, incorporating the start codon, 5′-AGTAAATTCTAGAATGCCTGGATTTGTGACTGTAGGCG-3′ and reverse primer “TcdB-XhoI-rev” that anneals up to the end of the stop codon, 5′-ACATATCTCGAGCTATTCACTAATCACTAAT-3′. Restriction sites included in the primer sequences are underlined. PCRs were performed in an Eppendorf^®^ Mastercycler^®^ (Stevenage, UK). PCRs typically consisted of an initial denaturation at 98 °C for 5 min followed by 35 cycles: denaturation at 98 °C for 30 s, annealing at the appropriate temperature for the primers for 30 s and extension at 72 °C for 30 s per 1 kb, and a final extension at 72 °C for 2 min. The amplicons generated were digested with SacI and XbaI or XbaI and XhoI and ligated into the SacI-XbaI site or XbaI-XhoI sites of dephosphorylated pMTL84123 [51,52]. Constructs were transformed into Top10 competent cells (ThermoFisher Scientific, Loughborough, UK) with selection on Chloramphenicol.

Sequence-verified constructs were designated pMTL84123_0873 and pMTL84123_TcdB and plasmid DNA used to transform *E. coli* sExpress strain with selection on Chloramphenicol and then introduced into NTCD T7 by conjugal transfer with selection on Thiamphenicol as described by Woods et al. (2019) [49]. Verified transconjugants were designated T7-0873 and T7-TcdB.

### 2.8. Western Immunoblotting of Whole Cells and Supernatants to Confirm Antigen Expression and Localisation

Strains of T7 were grown in 10 mL BHIS to *A*_600 nm_ 1.0 and enumerated by plating 1:10 serial dilutions on BHIS CC agar and enumerating CFU to confirm equivalent cell numbers. Whole cell lysates (WCL) were prepared by centrifuging 1 mL of culture and resuspending the pellet in 50 µL PBS plus 50 µL 2× Laemmli sample buffer. Supernatants were prepared by centrifuging 10 mL culture, discarding the cell pellet and filter sterilising through a 0.22 µM filter and then concentrating 10-fold using a Vivaspin 20 column with a 10 kDa cut-off (GE Healthcare Life Sciences, Amersham, UK). Equal volume of 2× Laemmli sample buffer was added prior to loading on a gel.

Western immunoblotting was performed with 5% dry-milk (*w*/*v*) (Sigma, Merck Group, Feltham, UK) in Tris buffered saline containing 0.01% Tween (*v*/*v*) (TBST) for blocking. All antibodies were diluted in 1% *w*/*v* dry-milk in TBST and TBST used for washes. Samples were fractionated by 10% (*w*/*v*) SDS-PAGE and then transferred to PVDF membranes using the Trans-Blot Turbo Transfer System (Bio-Rad, Hercules, CA, USA). Primary antibodies used were rabbit anti-CD0873 antibody (made by Antibody Production Services, Bedford, UK) (1:5000) and mouse anti-TcdB antibody (1:1000) (The Native Antigen Company, Oxford, UK). Secondary antibodies used were anti-rabbit IgG horseradish peroxidase (HRP) (1:1000) (Cell Signalling Technology, Danvers, MA, USA) and anti-mouse IgG HRP (1:1000) (Cell Signalling Technology, Danvers, MA, USA). Protein bands were detected using Enhanced Chemiluminescence (ECL) Western Blotting Detection reagent (Cytiva, Marlborough, MA, USA) and visualised using the Odyssey^®^ Fc imaging system (LI-COR Biosciences, Lincoln, NE, USA).

### 2.9. Immunisations of Hamsters

Female Golden Syrian hamsters aged 10–12 weeks, weighing around 120 g, were purchased from Janvier Labs (Le Genest-Saint-Isle, France) and housed in pairs in individually ventilated cages. Hamsters were randomly divided into 4 groups, each *n* = 4. After acclimatisation for 1 week, the placebo control group was given 100 µL PBS and the 3 experimental groups were given 100 µL suspension containing 10^6^ spores of NTCD T7, T7-0873 or T7-TcdB orogastrically. Dosing was performed on days 1, 15 and 30 and hamsters euthanised on day 45. Blood was collected by cardiac puncture, left to clot overnight at 4 °C and serum harvested after centrifugation. The small intestine was placed in 5 mL ice-cold PBS containing SIGMAFAST™ protease inhibitors (Sigma, Merck Group, Feltham, UK), flushed with this suspension and then the supernatant collected after centrifugation. Sera and intestinal lavages were filter-sterilised and stored at −80 °C.

### 2.10. Direct ELISA to Measure Total sIgA in Intestinal Lavages

Hamster sIgA was detected using a Hamster sIgA ELISA kit (MyBioscource, San Diego, CA, USA) according to manufacturer’s instructions. Briefly, intestinal lavages, diluted 1:2 in 50 µL of sample diluent buffer were added to the plate provided, in triplicate, along with 100 µL of an HRP-conjugated reagent. After 60 min incubation, the plate was washed 4 times with the wash buffer provided. The chromogen reagent was added; then, 15 min later, the stop solution was added and the plate immediately read at *A*_450 nm_.

### 2.11. Indirect ELISA to Measure IgG Titre in Sera

Ninety-six-well Nunc MaxiSorp™ plates were coated with 100 µL purified recombinant proteins; CD0873 and TcdB-RBD at a concentration of 0.5 µg/mL in 0.2 M sodium bicarbonate, pH 9.4, and the proteins were left to adsorb onto the wells overnight at 4 °C. All wash stages consisted of 5 washes with 200 µL PBS containing 0.01% Tween (PBST). Wells were first blocked with 200 µL of 5% dry-milk (Sigma, Merck Group, Feltham, UK) in PBST for 2 h at room temperature, washed and then incubated overnight at 4 °C after addition of 100 µL serum diluted 1:10 in PBST, in triplicate. Wells were washed and then incubated for 2 h at room temperature in 100 µL goat anti-hamster IgG highly cross adsorbed-Biotin antibody (Sigma, Merck Group, Feltham, UK) diluted 1:20,000 in PBST, washed again and then incubated for 2 h in Streptavidin-HRP (R&D Systems, Minneapolis, MN, USA) diluted 1:200 in PBST. After a final wash, 100 µL TMB substrate (Sigma, Merck Group, Feltham, UK) were added and the plate incubated at room temperature for 15 min. The reaction was stopped by addition of 100 µL H_2_SO_4_ to each well and absorbance values read at *A*_450 nm_ using the CLARIOstar^Plus^ (BMG Labtech, Ortenberg, Germany) Plate Reader.

### 2.12. Adherence Blocking Assay

Caco-2 cells were cultured in Dulbecco’s Modified Eagles Medium (DMEM) (Gibco™ ThermoFisher Scientific, Loughborough, UK) supplemented with 4.5 g/L D-glucose, 584 mg/L L-glutamine, 25 mM HEPES, 10% *v*/*v* FBS (Sigma, Merck Group, Feltham, UK) and penicillin/streptomycin (Sigma, Merck Group, Feltham, UK) in a humidified 5% CO_2_ atmosphere at 37 °C. Cells were seeded at 5 × 10^4^ cells per well in 24-well tissue culture plates (Corning, New York, NY, USA). Monolayers were used 14 days after seeding with media changed every 2–3 days. The culture media were further changed 24 h prior to conducting the assay. The inoculum was prepared by standardising the optical density (OD) of a 10 mL overnight broth culture of NTCD strains to *A*_600 nm_ 0.6, centrifuging and washing the cells with PBS and resuspending the pellet in non-supplemented DMEM. Caco-2 monolayers were infected at a multiplicity of infection (MOI) of 1:5 and 1:20 in triplicate. To confirm the MOIs, 1:10 serial dilutions of the cell suspension in PBS were plated onto BHIS CC for enumeration of CFUs. The adherence assay was performed under anaerobic conditions at 37 °C. Then, 50 µL of serum or intestinal lavage diluted 1:5 and 1:2 respectively in non-supplemented DMEM were added to 50 µL of the bacterial cell suspension and incubated for 1 h. This mixture was then added to Caco-2 cells in triplicate following removal of the media in the wells and adding 400 µL of fresh non-supplemented media. After 2 h of incubation, non-adherent bacteria were removed by pipetting and adherent bacteria harvested as follows. Caco-2 cells were washed 3 times with PBS, incubated in 200 µL 1× trypsin-EDTA (Sigma, Merck Group, Feltham, UK) to detach them from the wells and then resuspended in 300 µL supplemented DMEM. Neat to 10^−3^ dilutions of cells in BHIS broth were plated on BHIS CC agar in triplicate and CFU enumerated the following day.

### 2.13. Toxin Neutralisation Assay

Vero cells were seeded at 1 × 10^4^ per well in a 96-well plate in 50 µL phenol red-free DMEM (Gibco™-ThermoFisher Scientific) supplemented as described above and incubated for 18–20 h at 37 °C in 5% CO_2_. Serum and intestinal fluid were serially diluted 2-fold (1:4 to 1:512) in serum-free, phenol red-free DMEM and mixed with either an equal volume of TcdA at 100 ng/mL, TcdB at 100 pg/mL (Public Health England, London, UK), or media only for 1 h at 37 °C. Neat and serial dilutions of serum or intestinal fluid–toxin mixtures were added to Vero cells in triplicate to give a total well volume of 100 µL, and plates were incubated for 22 h at 37 °C. The final concentration of TcdA and TcdB was 25 ng/mL and 25 pg/mL, respectively. Cell rounding was investigated by microscopy.

### 2.14. Ethics Statement

Animal studies were devised using the Experimental Design Assistant (EDA) online tool and conducted in strict accordance with the requirements of the Animals Scientific Procedure Act 1986. Prior approval for these procedures was granted by the University of Nottingham Animal Welfare and Ethical Review Body and by the United Kingdom Home Office and the work conducted under project license, PP2643068. Animals were euthanised by CO_2_ inhalation followed by cervical dislocation to minimize suffering.

### 2.15. Statistical Analysis

All non-normally distributed data from this study were analysed by a nonparametric ANOVA (Kruskal–Wallis) with Dunn’s uncorrected comparison. The normally distributed data for protein expression were analysed by one-way ANOVA with Dunnett’s multiple comparisons. All statistical tests were performed using GraphPad version 7 (San Diego, CA, USA), and *p* values of less than 0.05 were considered to indicate statistical significance.

## 3. Results

### 3.1. Characterisation of NTCD Chassis Strain, T7

NTCD T7 was chosen, as it is commonly found in asymptomatic individuals and is considered safe [46]. Additionally, it was one of the three NTCD strains characterised for its natural protective effects against CDI in promoting colonisation resistance in hamsters pre-treated with antibiotics [45,46].

#### 3.1.1. NTCD T7 Lacks paLoc

The pathogenicity locus, paLoc, characteristic of TCD strains is 19.6 kb and encodes TcdR, TcdB, TcdE, TcdA and TcdC. NTCD strains typically possess a conserved 115 bp non-coding region in place of paLoc. Although the evolutionary history of NTCD remains unclear as to whether non-toxigenic lineages once produced toxins or whether toxin-producing lineages evolved from non-toxigenic ancestors [53], the initial discovery of the 115 bp sequence in NTCD strain 1351 led to its designation as the “paLoc integration site” [54]. Illumina sequencing of genomic DNA of NTCD T7 was conducted, and 144 contigs were generated. The whole genome shotgun was deposited in GenBank as a BioProject under Accession PRJNA826427. The integration site from strain 1351 was used as a query sequence in BLASTn analysis of the T7 shotgun. The site was found on contig 106 and demonstrated 95% nucleotide identity with that of strain 1351. Its sequence is shown in Figure 1A. Further BLASTn analyses using paLoc genes and paLoc flanking sequence of the TCD reference strain 630 as query sequences confirmed the absence of the entire 19.6 kb paLoc in NTCD T7 and enabled mapping of the integration site relative to 630. The site mapped from 903 nucleotides upstream of *tcdR* to 231 nucleotides downstream of *tcdC* (Figure 1B). The region flanking the integration site demonstrated high nucleotide identity with the region flanking paLoc in strain 630, with upstream gene CD630_06580 (*cdu1*) sharing 99% nucleotide identity and downstream gene CD630_06641 (*cdd1*) sharing 100% identity.

#### 3.1.2. NTCD T7 Encodes Non-Toxin Antigens

One of the benefits of deploying NTCD as a vaccine chassis in addition to its natural attenuation by the absence of paLoc is the presence of non-toxin antigens in common with TCD which are expected to afford some level of protection against infection by this organism. Homologues of non-toxin protein antigens that have shown promising results when preclinically tested were searched for in the whole-genome shotgun of NTCD T7 using genes of TCD strain 630 and their corresponding translated sequences as query sequences. Table 1 shows that the percentage nucleotide and amino acid identities of these vaccine candidates are 99%–100%, except for SLpA. Promising non-protein antigens such as oligosaccharides [55], lipoteichoic acids, and carbohydrate polymers [56,57] are also common in the cell walls of NTCD and TCD but were not included in this analysis due to the complexity of numerous genes required for their biosynthesis.

#### 3.1.3. Spores of NTCD T7 Germinate in the Distal Small Intestine

The relative abundance of primary bile salts (germinants) and secondary bile salts is regulated by the microbiota, which in turn is affected by the action of antibiotics. In order for an NTCD vaccine administered in spore form to induce an immune response, it must first germinate into metabolically active cells at or before arriving at the ileum to successfully deliver antigens to the main immune inductive site. Ultimately, our intention is to administer vaccines to healthy individuals not on antibiotics who are at risk of contracting CDI, and thus, it is also necessary that germination occurs in the absence of antibiotics.

To test where spores germinate in the small intestine and whether germination occurs in the absence of antibiotics in hamsters, an ex vivo study was conducted based on that described by Kochan et al., (2018) [16]. Briefly the proximal, mid and distal portions (each representing a third of the small intestine) were separately combined from 6 hamsters (not pre-treated with antibiotics) and the intestinal contents extracted. Spores were incubated in the 3 pooled fluids for 1 h anaerobically then plated onto BHIS agar containing selective supplements and Kanamycin to eliminate any host commensal bacteria present. This media only supports the growth of germinated spores due to the lack of bile salts. After anaerobic incubation, the following day CFU were enumerated and compared to the positive control sample, i.e., spores from the same stock germinated in BHIS plus 0.1% taurocholate which was defined as 100% germination.

Over 10% of spores germinated in the small intestine relative to the positive control: 9.9% in distal, 0.2% in proximal and 0.5% in mid small intestine (Figure 2). Our findings for location of germination support those of Kochan et al., (2018), who reported in mice that germination occurred significantly more efficiently in the ilea compared to the duodenum and other parts of the GI tract tested [16]. Our observation that germination occurred in the absence of antibiotics also supports conclusions reached by Kochan et al. (2018), that in mice, *C. difficile* spores are able to germinate in ileal contents independent of antibiotic treatment (by overcoming the effects of inhibitory bile salt chenooxycholate with the combined activities of taurocholate, glycine and calcium) [16]. Moreover, several studies from the Young and Theriot groups have shown that while germination may be enhanced after antibiotic treatment, it is always supported in murine ileal contents [17,21,65]. From our analyses of the genome and germination of NTCD T7, this strain was deemed appropriate for further development as a vaccine chassis in which to engineer antigens.

### 3.2. Recombinant Overexpression of Antigens, CD0873 and TcdB-RBD, in NTCD T7

The adhesin CD0873 was chosen, as it has been shown to induce antibodies with colonisation-blocking properties. Wright et al. (2008) first discovered that this antigen is sero-reactive in CDI patients [66]. Subsequent structural and functional analyses revealed that CD0873 is a surface-exposed lipoprotein involved in adhesion to Caco-2 cells [67]. Recombinant CD0873 formulated with adjuvant, given by intraperitoneal (ip) injection, was found to be effective in mice in preventing long-term gut colonisation with TCD [24]. Supporting these findings, we showed that recombinant CD0873 given orally to hamsters induced strong intestinal sIgA and serum IgG responses with colonisation-blocking properties that afforded partial protection from infection by a hypervirulent strain of *C. difficile* [25]. For the toxin-based antigen, the RBD of TcdB was chosen, as this was previously shown to be immunogenic when administered ip in encapsulated nanoparticles to mice [48].

#### 3.2.1. Cloning of Antigens in pMTL84123 for Expression in NTCD T7

The gene encoding CD0873 and the region encoding the RBD of TcdB in the chromosome of TCD strain 630 were cloned into the *C. difficile* expression vector pMTL84123 [52]. This plasmid carries a low copy number *E. coli* replicon and the conjugation transfer sequence, the *C. difficile* pCD6 replicon, the CAT selectable marker and the *Clostridium*
*sporogenes fdx* promoter for constitutive, strong expression of the cloned gene [51]. For CD0873, the putative native promoter region was included, and for TcdB-RBD, a start codon was included. The plasmids generated, pMTL84123_0873 and pMTL84123_TcdB, were used to transform *E. coli* sExpress [49] with selection on Chloramphenicol, which were then introduced into NTCD T7 by conjugal transfer with selection on Thiamphenicol. Transconjugants were verified by PCR amplification and sequencing. Verified strains were designated T7-0873 and T7-TcdB.

#### 3.2.2. Confirmation of Expression of Antigens in Strains T7-0873 and T7-TcdB

Western immunoblots were conducted for fractionated whole cell lysates (WCL) and concentrated supernatants with anti-CD0873 antibody and anti-TcdB antibody. Samples were prepared from early stationary phase cultures and CFU enumerated to verify that equivalent cell numbers were used for each strain. Since CD0873 is present in the genome of NTCD T7, the native level of expression is expected to be detected in all strains and enhanced in T7-0873 due to recombinant overexpression. CD0873 was detected in all strains, both intracellularly and in the supernatant. The detection of CD0873 in the supernatant was expected since this lipoprotein is known to be extracellular as evidenced by its presence in culture filtrates of TCD strains, including strain 630 [68]. A significantly greater level of CD0873 was detected in WCLs and supernatants of T7-0873, as expected (*p* = 0.0104, *p* < 0.0001, respectively), but surprisingly also in strain T7-TcdB (*p* = 0.032, *p* < 0.0001, respectively) (Figure 3).

TcdB-RBD was detected only in T7-TcdB and in the WCL mainly with less in the supernatant (Figure 3). The region encoding TcdB-RBD in construct pMTL84123_TcdB lacks a signal peptide sequence for translocation across the cytoplasmic membrane, unlike CD0873, which naturally harbours a signal peptide for both translocation and lipidation [69]. The TcdB-RBD expressed by T7-TcdB should therefore not exit the cytoplasm unless as a result of cell lysis.

### 3.3. Immunisation Regimen and In Vitro Cell Models to Assess Antibody Functionalities

A suspension of 1 × 10^6^ spores of each strain, T7-0873, T7-TcdB and T7, were used to immunise hamsters orally three times, 2 weeks apart. Animals were humanely euthanised 2 weeks after the last immunisation. The small intestine was removed and flushed with 5 mL of PBS containing protease inhibitors, diluting the intestinal fluid about 50-fold. Sera were obtained by allowing blood harvested by cardiac puncture to clot overnight. Following centrifugation and filter-sterilisation of supernatants, intestinal lavages and sera were frozen. In order to investigate vaccinated fluids for functional antibodies, the Caco-2 human intestinal epithelial cell line was used to assess adherence blocking, as this is a widely used model to investigate colonisation by *C. difficile*, and Vero cells (African Green Monkey kidney epithelial cell line) used to assess toxin neutralisation, as this model is routinely used for *C. difficile* cytotoxicity assays [25].

#### 3.3.1. Intestinal Immune Responses in Vaccinated Hamsters

First, to measure the titre of sIgA in intestinal lavages, since no anti-hamster sIgA antibody is commercially available, an anti-hamster sIgA antibody ELISA kit (MyBioSource) was used. Intestinal lavages were diluted 1:2. Only hamsters immunised with T7-0873 induced a significant titre of sIgA (*p* = 0.0103) (Figure 4A) with a corresponding significant reduction in the binding of TCD 630 to Caco-2 cells, *p* = 0.0123 (Figure 4B). All immunised hamsters were further tested for the presence of toxin-neutralising antibodies, but none were found, including hamsters administered T7-TcdB.

#### 3.3.2. Systemic Immune Responses in Vaccinated Hamsters

To further test for the presence of antigen-specific IgG in vaccinated sera, 96-well plates were coated in recombinant protein antigen, either CD0873 or TcdB-RBD as previously described [25,47]. After blocking, sera diluted 1:10 were added and bound antibody detected using goat anti-hamster IgG Biotin-labelled secondary antibody and Streptavidin-HRP.

Significant titres of antigen-specific IgG were detected in hamsters administered T7-0873 (*p* = 0.0026) (Figure 5A) and T7-TcdB (*p* = 0.0412) (Figure 5B). High cross reactivity of serum IgG was observed in T7-0873-vaccinated hamsters to recombinant TcdB-RBD (*p* = 0.0022) (Figure 5B). Sera diluted 1:5 were then tested for adherence blocking activities. The sera of both groups exhibited significant reduction in the adhesion of TCD 630 cells to Caco-2 monolayers (*p* = 0.027, *p* = 0.0038 respectively) (Figure 5C). The sera of animals were also tested for the presence of toxin-neutralising antibodies, but none were found. To conclude, for T7-0873-vaccinated hamsters, serum IgG responses mirrored the sIgA responses seen in the intestinal fluid. For hamsters immunised with T7-TcdB, humoral responses were observed in sera only.

To summarise, we show that NTCD T7 is a promising vaccine chassis. It is capable of germinating in the small intestine, mainly in the ileum, in the absence of antibiotics and harbours all non-toxin protein antigens common with TCD that have previously shown promise in preclinical studies. Importantly, we show that overexpression of the colonisation factor, CD0873, by NTCD T7 successfully induces antibodies which reduce the adhesion of TCD 630 to epithelial cells. Moreover, this was observed in intestinal fluid that was heavily diluted (1:100) and in diluted serum (1:5). These encouraging results warrant a challenge study to investigate the protective efficacy of recombinant NTCD against CDI. In its current form, there is the possibility that T7-0873 will colonise the host, and thus, any protective effects observed against CDI could be partly attributed to colonisation resistance rather than exclusively to the promotion of a humoral response. The next steps now are to modify the chromosome of NTCD T7 to prevent its persistence in the host, increase its germination efficiency and express elevated levels of CD0873 and immunogenic toxin domains to potentially block all stages of pathogenesis.

## 4. Discussion

With *C. difficile* posing a critical global health threat and with the failure of two intramuscular toxoid vaccines to prevent primary CDI in phase III trials [37,38], deploying the oral route is emerging as an attractive solution to induce vital local protection. Developing oral vaccines of the safe, subunit type is challenging due to their degradation in the stomach. One way to overcome this bottleneck is to exploit a gut commensal spore-former as a vaccine chassis for the natural delivery of intact engineered antigens to the small intestine.

Wild-type NTCD has already been tested as a biotherapeutic in patients experiencing CDI and is an effective preventive for recurrent CDI by promoting colonisation resistance [31,43]. Encouragingly, NTCD strain M3 has now been FDA approved for advancement to Phase III. In addition to the whole cell approach, the membrane fraction of wild-type NTCD given intrarectally with cholera toxin as adjuvant affords significant protection from CDI in animal models [70]. A 99% decrease in TCD cells was observed in the faeces of challenged vaccinated mice, and a 30% increase in survival was demonstrated in challenged vaccinated hamsters compared to control groups administered cholera toxin only [70]. Protection induced by the membrane fraction of NTCD against TCD infection can be explained by common antigens shared between these two organisms. A comparison of antigens (both vegetative and spore associated) in this study revealed extremely high conservation between NTCD T7 and TCD 630 (Table 1).

Regarding the safety of NTCD, testing in animals and humans has confirmed this organism to be safe. However, a concern was raised that in clinical use, NTCD strains may acquire the paLoc and become pathogenic. Brouwer et al. (2013) showed in vitro that the mating of TCD with NTCD strains resulted in the transfer of paLoc, albeit at low frequency [71]. The likelihood of horizontal transfer of paLoc occurring in humans administered NTCD is unknown but would inevitably require an abundance of both organisms residing in close proximity in the GI tract. In our study, NTCD was deployed for targeted delivery of engineered antigens to the ileum in non-antibiotic treated animals, rather than as a biotherapeutic for antibiotic-treated animals. The ultimate goal will be for recombinant NTCD to persist in the host for the minimum length of time needed to deliver antigens then lyse, rather than proceed to colonise the host; thus, the chance of acquiring paLoc would be negligible.

For deploying NTCD as a vaccine, we first determined whether spores of NTCD T7 could germinate in the small intestine, which is essential for the engineered antigens to be expressed and released. Not all primary bile salts in the ileum promote germination. The bile acid chenodeoxycholic acid is secreted into the duodenum, which like cholic acid, is conjugated with taurine or glycine, and then deconjugated in the ileum to generate chenodeoxycholate [19]. Chenodeoxycholate is a germination inhibitor, and this is converted in the colon to the secondary bile acid lithocholic acid, another inhibitor [19]. Since the balance of bile salts is regulated by the microbiota which in turn is affected by the action of antibiotics, we tested whether spores of NTCD T7 can germinate in non-antibiotic treated hamsters. Our findings confirmed that germination occurs in the small intestine of untreated hamsters and mainly in the ileal fluid albeit at a low rate. We would anticipate the rate of germination to be higher in the natural in vivo environment than in ex vivo conditions.

Building on a previous study by Wang et al. (2018) [44], we aimed to determine whether NTCD could be engineered to induce antibodies with adherence blocking properties against *C. difficile* to potentially prevent the first stage of pathogenesis. One of the many benefits of deploying NTCD is the likely folding and display of antigens in a manner akin to TCD, which will potentially induce antibodies with high avidity. We chose the colonisation factor CD0873, as recombinant forms of this protein have previously been shown to induce colonisation-blocking antibodies when given ip to mice with adjuvant, or orally to hamsters [24,25]. For the toxin-based antigen, the RBD of TcdB was chosen, as this is known to be immunogenic in mice [48]. T7 was chosen as a chassis strain to engineer chosen antigens, as it has previously been tested as the wild-type strain in hamsters for protection against CDI [45,46].

T7-0873 induced a significant humoral response intestinally as well as systemically with antibodies from both fluids significantly reducing the adherence of cells of TCD 630 to Caco-2 cells. T7-TcdB also induced antibodies with significant adherence blocking activity but in serum only and without toxin-neutralising activity. One possible reason for the difference in ability of these antigens to induce intestinal responses is that CD0873 may be more available in the host than TcdB-RBD. CD0873 is found extracellularly, as seen by its abundance in the supernatant, whereas TcdB-RBD is expressed cytosolically (Figure 3) and would therefore rely mainly on cell lysis for its release. This would likely mean that less TcdB-RBD is available for uptake by enterocytes such as M cells which sample antigens for delivery to the GALT. Another reason may be that CD0873 is simply more immunogenic in hamsters than toxins. Wang et al. (2018) reported a lower titre of toxin neutralising serum antibodies in hamsters compared to mice immunised orally with NTCD_mTcd138 [44]. In hamsters, partial protection was observed against CDI as determined by percentage of animals that survived and did not display diarrhoea whereas mice were fully protected [44]. An earlier study by Wang et al. (2015) testing a recombinant version of their toxin fusion protein administered parenterally also reported differences in potencies between hamsters and mice. Again neutralising titres against both toxins were lower in the sera of vaccinated hamsters compared to vaccinated mice, which was reflected by partial protection against infection with a hypervirulent strain in hamsters, compared with full protection from any appreciable signs of disease in mice [72]. The clear differences in the ability of the two rodents to generate toxin-neutralising antibodies may partly explain their profound differences in susceptibility to CDI.

Although T7-TcdB did not induce antibodies with toxin-neutralising activity, the observation that this strain generated adherence blocking antibodies systemically may be due to their potency, given the relatively low titre (compared to anti-CD0873 IgG induced by T7-0873). Alternatively, a combination of anti-TcdB antibodies and anti-CD0873 antibodies worked together synergistically. T7-TcdB expressed significantly more CD0873 than the wild-type strain (Figure 3). This may be due to polar effects from the overexpression of TcdB-RBD.

Another unexpected finding from our study was that systemic IgG induced by T7-0873 was not only specific for CD0873 (Figure 5A) but cross-reacted with TcdB-RBD (Figure 5B). A high degree of cross-reactivity was also reported by Wang et al. (2018), who measured the titre of anti-FliCD antibodies in animals immunised with NTCD_mTcd138 compared with the wild-type strain. The data suggest that in mice, a greater titre of anti-FliCD IgG was found in sera of the NTCD_mTcd138-vaccinated group, furthermore anti-FliCD serum IgA and anti-FliCD faecal IgG and IgA were only detected in this group. In hamsters where only IgG responses could be measured, similar observations were made in that anti-FliCD IgG was found in sera for both groups but only in the faeces of hamsters vaccinated with NTCD_mTcd138 [44]. Clearly, the recombinant strain expressing toxin domains was effective in inducing antibodies not only specific for the toxin chimera but for unrelated native flagellar proteins, and it would be interesting to determine whether FliC and/or FliD expression levels were higher in the recombinant strain compared to the wild-type. Hong and colleagues also reported cross-reactivity with their *Bacillus subtilis* spore vaccine engineered to express a carboxy-terminal segment of TcdA on the spore surface [73]. In orally vaccinated hamsters, antibodies were not only specific to the TcdA fragment but cross-reactive with unrelated proteins expressed on the surface of both vegetative and spore forms of *C. difficile* despite no meaningful homology in amino acid sequences [73]. The findings taken together suggest that immunogenic toxin domains administered orally have the potential to induce antibodies that neutralise toxins (as observed in Wang’s studies and by Hong and co-workers), as well as antibodies that block adhesion, as observed in our study, which may be aided by cross reactivity to bacterial surface factors.

Further work is now needed to generate an improved vaccine chassis with deletion of genes that will prevent persistence in the host. Other chromosomal modifications are needed to increase the germination rate and integrate antigen-encoding genes in the chromosome for stability, which can be achieved without the inclusion of antibiotic resistance markers. A stronger constitutive promoter may also be necessary as well as increasing the number of immunisations to deliver a sufficient quantity of antigen, in particular toxin domains to successfully induce toxin-neutralising antibodies. The combination of CD0873 and immunogenic domains of TcdB and TcdA may be sufficient to block colonisation and neutralise the toxins but if required other antigens can be included using a multiple chromosomal integration approach or by generating a panel of recombinant strains and administering a spore cocktail to target all stages of pathogenesis.

The benefits of using spores as a vaccine platform are manifold. Spores are extremely hardy and therefore have an “indefinite” shelf life. Being thermostable, they do not require cold-chain for storage or transport and are inexpensive to manufacture. The spore platform further benefits from the generic advantages of oral vaccines over parenteral vaccines such as ease of administration and increased compliance, such as painless delivery and lower cost, as no needles or syringes or trained workers are required. Additionally, oral vaccines are safer than injected vaccines, as the risks of needle stick injury and cross contamination are avoided. With NTCD proven to be safe in humans [43], recombinant forms of this organism may offer an attractive solution not only to target *C. difficile* but other pathogens for which vaccines are needed. The thermostability of the spore platform is particularly practical for global immunisation, an area that has attracted great interest since the COVID-19 pandemic.

## Figures and Tables

**Figure 1 pharmaceutics-14-01086-f001:**
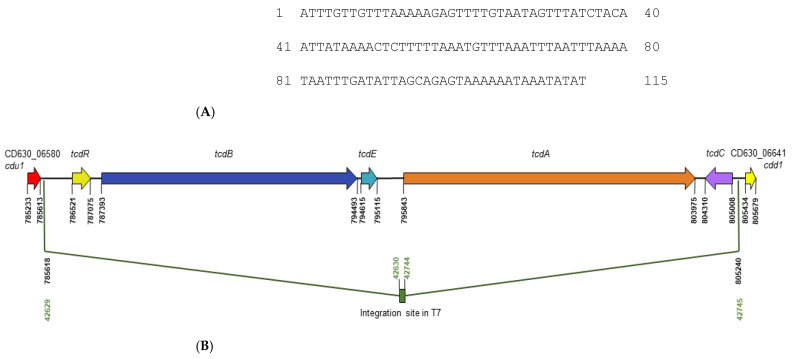
The identification of the 115 bp non-coding integration site in the genome of non-toxigenic *C. difficile* (NTCD) strain T7. (**A**) The nucleotide sequence of the integration site in T7 (**B**) Mapping of the integration site in T7 relative to the genome of the TCD reference strain, 630. The site mapped to intergenic sequence 903 nucleotides upstream of *tcdR* and 231 nucleotides downstream of *tcdC*.

**Figure 2 pharmaceutics-14-01086-f002:**
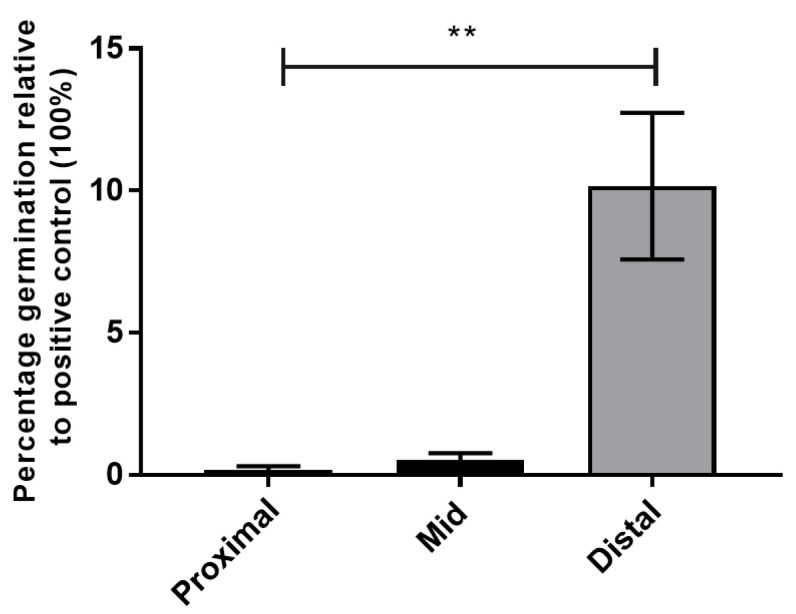
Spores of NTCD T7 germinate in the small intestine of hamsters. Spores were incubated in the contents of fluid taken from proximal, mid and distal portions of non-antibiotic-treated hamsters (*n* = 6) for 1 h at 37 °C anaerobically. Data are presented as percent germination, determined by: (CFU from germinated spores pre-incubated in intestinal fluid/CFU from germinated spores pre-incubated in 0.1% taurocholate) × 100. Error bars represent the standard deviation (SD). The data were analysed by nonparametric ANOVA (Kruskal–Wallis) with Dunn’s uncorrected comparison. Statistical difference *p* value: ** *p* = 0.0017.

**Figure 3 pharmaceutics-14-01086-f003:**
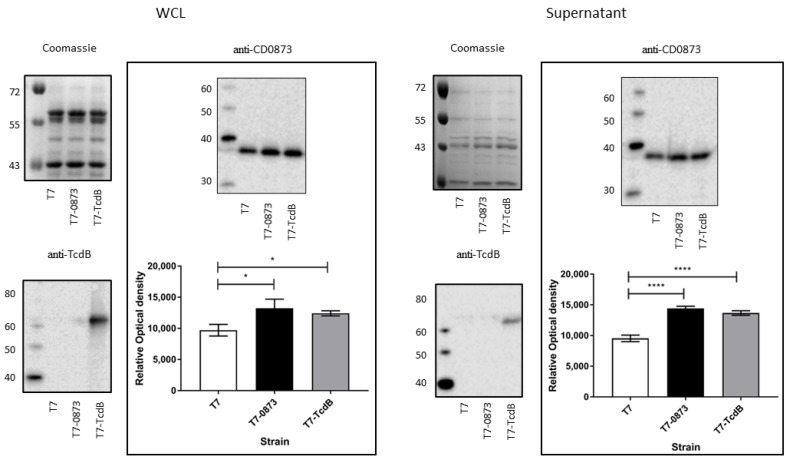
Detection of the expression of antigens in recombinant strains of T7 by SDS-PAGE (10% acrylamide) and Western immunoblotting. Whole cell lysates (WCL) and supernatants of strains T7, T7-0873 and T7-TcdB were harvested from BHIS broth cultures at early stationary phase, *A*_600 nm_ 1.0. Replica gels for both WCLs and supernatants were stained by Coomassie Blue to confirm equal loading of samples. Western immunoblots with anti-TcdB antibody detected TcdB-RBD mainly intracellularly in T7-TcdB. Western immunoblots with anti-CD0873 antibody detected CD0873 in WCLs and supernatants of all strains with significantly enhanced levels in both recombinant strains. Bands were detected by the addition of the appropriate HRP-conjugated secondary antibody and ECL Western Blotting Detection Reagent. Densitometry was carried out in ImageJ. Error bars represent the standard deviation (SD). Data were analysed by a one-way ANOVA with Dunnett’s multiple comparison. Statistical difference *p* values: * *p* < 0.05, **** *p* < 0.0001.

**Figure 4 pharmaceutics-14-01086-f004:**
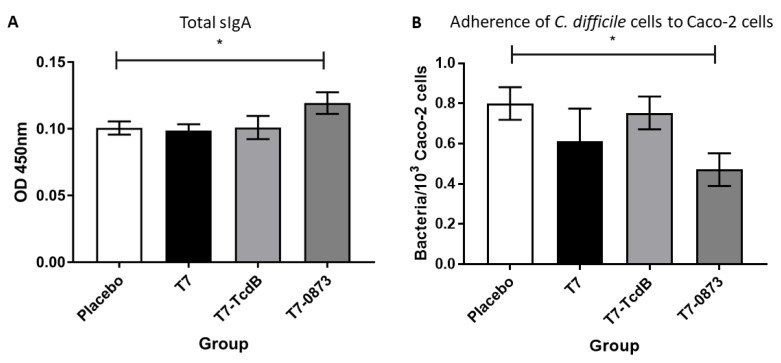
Intestinal immune responses in hamsters 2 weeks after oral immunisations with 10^6^ spores of strains T7, T7-TcdB and T7-0873: 3 doses, 2 weeks apart. (**A**) Total sIgA in intestinal lavages diluted 1:2 was measured by an anti-hamster sIgA ELISA kit (MyBioSource). (**B**) Adherence of *C. difficile* strain 630 to Caco-2 monolayers after the bacterial cells were pre-incubated with intestinal lavages diluted 1:2. Cell binding was measured by enumerating CFU from washed Caco-2 cells. Error bars represent the standard deviation (SD). The data were analysed by nonparametric ANOVA (Kruskal–Wallis) with Dunn’s uncorrected comparison. Statistical difference *p* value: * *p* < 0.05.

**Figure 5 pharmaceutics-14-01086-f005:**
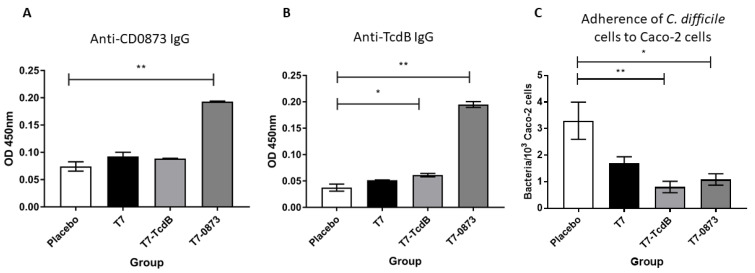
Systemic immune responses in hamsters 2 weeks after oral immunisations with 10^6^ spores of strains T7, T7-TcdB and T7-0873: 3 doses, 2 weeks apart. Sera were diluted 1:10 and quantified for antigen-specific IgG by indirect ELISA: CD0873 (**A**) and TcdB-RBD (**B**). Goat anti-hamster IgG highly cross-adsorbed Biotin antibody (1:20,000) and Streptavidin-HRP (1:200) were used for detection. (**C**) The effect of sera diluted 1:5 in reducing the binding of *C. difficile* strain 630 to Caco-2 cells. Error bars represent the standard deviation (SD). The data were analysed by nonparametric ANOVA (Kruskal–Wallis) with Dunn’s uncorrected comparison. Statistical difference *p* values: * *p* < 0.05, ** *p* < 0.01.

**Table 1 pharmaceutics-14-01086-t001:** Identification of homologues of TCD antigens in NTCD T7 that have shown some protection from either colonisation or infection in animal models. The percentage nucleotide and amino acid identity of these antigens with TCD strain 630 is shown.

Antigen	Annotated Function	Nucleotide Identity	Amino Acid Identity	Source
Cwp84	Cell wall-binding cysteine protease	2396/2412 (99%)	802/803 (99%)	[58,59]
GroEL	Heat shock protein	1625/1629 (99%)	541/542 (99%)	[60]
CD0873	ABC transporter substrate-binding protein. Adhesin	1022/1023 (99%)	340/340 (100%)	[24,25]
SLpA	S-layer precursor protein	825/1026 (80%) out of 2160 bp	427/733 (58%)	[61]
FlicC	Flagellin	873/873 (100%)	290/290 (100%)	[62]
FliD	Flagellin cap protein (tested in combination with flagellar preparation)	1524/1524 (100%)	507/507 (100%)	[63]
CdeC	Spore protein. (Exosporium morphogenetic protein)	1213/1218 (99%)	403/405 (99%)	[64]
CdeM	Spore protein. (Exosporium morphogenetic protein)	482/483 (99%)	160/160 (100%)	[64]

## Data Availability

The whole genome shotgun of NTCD strain T7 was deposited in GenBank as a BioProject under Accession PRJNA826427.

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
