# Peer review of "Towards Development of a Non-Toxigenic Clostridioides difficile Oral Spore Vaccine against Toxigenic C. difficile"

_pharmaceutics, 2022, doi:10.3390/pharmaceutics14051086_

Round 1
Reviewer 1 Report
In this manuscript, the authors evaluated an oral spore vaccine against Clostridioides difficile deploying a gut commensal for natural delivery of engineered antigens to the small intestine. This manuscript is overall a well controlled study.
- The introduction section is too long, refine the statement
- add the country and company for each regent
- Lots of introduction and discussion sentences were used in the results section. I think these sentences are useless and will affect the expression of the result. Remove these sentences to the discussion section.
Author Response
The authors thank Reviewer 1 for their thorough review of our manuscript and helpful comments.
- Point 1: The introduction section is too long, refine the statement
- Response 1: The introduction has been reduced by a third and now provides a more succinct background and rationale for the study.
- Point 2: add the country and company for each regent
- Response 2: The country and company for the reagents in Methods have now been included
- Point 3: Lots of introduction and discussion sentences were used in the results section. I think these sentences are useless and will affect the expression of the result. Remove these sentences to the discussion section.
- Rseponse 3: Information in the Results section has been moved to the Discussion where appropriate which allows stronger focus on each result per se in the Results section and a fuller application of the literature to these results in the Discussion section.
Reviewer 2 Report
The manuscript entitled “An oral spore vaccine against Clostridioides difficile deploying a gut commensal for natural delivery of engineered antigens to the small intestine” is very interesting and has a significant impact on the oral vaccine development. It needs some edits before acceptable for publication.
The major drawback of the manuscript was introduction. It was too lengthy and covered mostly reported data. Reduce the introduction and avoid redundancy.
In the abstract, the authors discussed “Two intra-muscular toxoid vaccines entered Phase 15 III trials but failed to prevent primary C. difficile infection (CDI)”. But there is no discussion about the treatment fail and why alternative route preferred?
Table 1 very poor quality and rewrite as per journal Table preparation format.
The statistical treatment of the data missing in Fig. 2.
Author Response
The authors thank Reviewer 2 for their very helpful comments and suggestions.
Point 1: The major drawback of the manuscript was introduction. It was too lengthy and covered mostly reported data. Reduce the introduction and avoid redundancy.
Response 1: The introduction section has been trimmed down by a third and now covers 2 pages of information relevant to the study. There is no redundant information provided now.
Point 2: In the abstract, the authors discussed “Two intra-muscular toxoid vaccines entered Phase 15 III trials but failed to prevent primary C. difficile infection (CDI)”. But there is no discussion about the treatment fail and why alternative route preferred?
Response 2: In the abstract, the sentence on the failure of two intramuscular vaccines is now followed by a more clear sentence explaining that alternatively immunising orally (as opposed to by injection) can directly target the ileum to generate immunoprotection in the colon:
“Alternatively, by immunising orally, the ileum (main immune inductive site) can be directly targeted to confer protection in the large intestine”
Point 3: Table 1 very poor quality and rewrite as per journal Table preparation format.
Response 3: Table 1 has been replaced using the Journal-recommended table format
Point 4: The statistical treatment of the data missing in Fig. 2.
Response 4: Statistical treatment of the data is now displayed in Figure 2 and the p value included in the figure legend.
Reviewer 3 Report
The publication "An oral spore vaccine against Clostridioides difficile deploying ..." is a fairly good manuscript that can be published with some corrections.
Below are my comments.
1. Please rewrite the title - it's too long and tiring.
2. The first part of the abstract should be rewritten, it adds little.
3. Can there be fewer keywords?
4. It would be worth considering a slight shortening of the introduction. Reading it, one gets lost.
5. What is Fig 1 about? Is it Panel A and B?
6. Table 1 is illegible.
7. I have no reservations about the discussion.
8. No clear summary - please add it.
9. 84 citations! It couldn't be more?
To sum up, the job is really nice, but a slight correction will not hurt it.
Author Response
The authors kindly thank Reviewer 3 for their thorough review and very helpful suggestions
Point 1: Please rewrite the title - it's too long and tiring.
Response 1: The title has been rewritten and now reads as “Towards development of a non-toxigenic Clostridioides difficile vaccine against toxigenic difficile”
Point 2: The first part of the abstract should be rewritten, it adds little.
Response 2: The first part of the abstract has been rewritten to succinctly describe the disease, the two failed intramuscular vaccines and the rationale for taking the oral vaccine approach adopted in this study.
Point 3. Can there be fewer keywords?
Response 3: The number of keywords has been reduced to those of greatest relevance for the study
Point 4. It would be worth considering a slight shortening of the introduction. Reading it, one gets lost.
Response 4: The Introduction has been shortened from 3 pages to 2 pages and is now more streamlined and succinct.
Point 5: What is Fig 1 about? Is it Panel A and B?
Response 5: Figure 1 displays the nucleotide sequence of the integration site and its match with the integration site of another strain (Panel A) and the positioning of the integration site in the genome (Panel B). The two panels for Figure 1 have now been boxed to make it more clear that these are two panels for the same figure.
Point 6: Table 1 is illegible.
Table 1 has been rewritten using the Journal’s recommended table format.
Point 7: I have no reservations about the discussion.
Response 7: The Discussion has essentially been kept as it was but with information transferred from Results where appropriate as recommended by Reviewer 1.
Point 8: No clear summary - please add it.
Response 8: A summary is included at the end of the Results section
Point 9: 84 citations! It couldn't be more?
Response 9: The citation list was admittedly length and now includes references of the greatest relevance (73 citations in total).
Round 2
Reviewer 2 Report
No further comments.